# EVADE: Exploring Vaccine Dissenting Discourse on Twitter

Shreya Ghosh[1], Prasenjit Mitra[2], Bernice L. Hausman[3]

College of Information Sciences and Technology[1,2]

Department of Humanities, College of Medicine[3]

The Pennsylvania State University, USA

shreya.cst@gmail.com[1], pum10@psu.edu[2], bhausman1@pennstatehealth.psu.edu[3]

## ABSTRACT

Social media plays a pivotal role in acquiring, exchanging and expressing public opinions and perceptions on a unprecedented scale in these pandemic times. In this paper, we develop an end-to-end knowledge extraction and management framework named as *EVADE*. This framework is used to automatically extract information consistent and inconsistent with scientific evidence regarding vaccination. Additionally, we seek to explore public opinion towards vaccination resistance proposing novel natural language processing methods. The knowledge extraction pipeline consists of three major modules, namely, knowledge-base construction, categorization of vaccine dissenting tweets, and effective analyses of discourses in those tweets effectively. Our major contributions lie in the fact that (i) the proposed knowledge extraction framework does not require huge amounts of labelled tweets of different categories and (ii) our module outperformed baselines by a significant margin of $\approx 8\%$ to $\approx 14\%$ in the classification tasks, and effectively analyze vaccine dissenting discourse.

## CCS CONCEPTS

• **Computing methodologies → Machine learning algorithms**; • **Information systems → Data mining**.

**ACM Reference Format:**

Shreya Ghosh[1], Prasenjit Mitra[2], Bernice L. Hausman[3]. 2022. EVADE: Exploring Vaccine Dissenting Discourse on Twitter. In *epiDAMIK 2022: 5th epiDAMIK ACM SIGKDD International Workshop on Epidemiology meets Data Mining and Knowledge Discovery, August 15, 2022, Washington, DC, USA.* ACM, New York, NY, USA, 10 pages.

## 1 INTRODUCTION

Twitter acts as platform to propagate information that is not based on scientific consensus. The term "misinformation" has been used widely without careful definition and precision. In order to avoid confusion, we define the following:

*Definition 1.1. Class_A* information (abbreviated as $Cl_A$) is used to denote facts that are accepted by most of the scientific community on the basis of evidence generated by rigorous scientific methods and subsequently peer-reviewed. Similarly, *Class_B* (abbreviated as $Cl_B$) information is all other information that is proposed without having the support of mainstream scientific consensus.

In this paper, we seek to (i) characterize $Cl_B$ information about vaccination on social media, and (ii) devise knowledge extraction techniques to identify vaccine dissenting discourse[1] and users involved in such dissent, as well as users who change their stance based on such discourse.

### Significance, Challenges, and Contributions

Supervised learning can be used to automatically detect $Cl_B$ and $Cl_A$ information consistent with scientific consensus from tweet discourse. Unfortunately, the availability of labelled data to train a supervised learning model is often insufficient. There is also temporal and location diversity along with other contexts, namely, external influence, political propaganda to name only a few, that impacts the public opinion in a significant way and the topic of discourse changes over time. Therefore, a fixed set of labels ("topics") of tweets does not seem realistic. We present a systematic knowledge extraction framework, which provides an overview of opinions expressed in tweets by analyzing the content (vaccine dissent and $Cl_B$ information) and analyzing the linguistic and semantic characteristics of tweets leveraging novel machine learning methods at different temporal scales. Analyzing heterogeneous data sources and extracting implicit information becomes more challenging when such data-instances are dynamic (as topics of discourse change based on varied influences) and voluminous. *Specifically, our problem is to classify vaccine dissenting tweets into different classes based on the reasoning given to support them (See Table 1).* To achieve that, we need to identify public stance ("against", "in favour" and "neutral") and sentiment ("negative", "positive" and "neutral") towards vaccination, followed by analysing vaccine dissenting tweets ("against" stance category and "negative" sentiment) to identify $Cl_B$ topics. However, efficient identification of public opinion in terms of stance (expressed in $Cl_B$ tweets) and sentiment is not straightforward, since there is no defined contextualization process to deal with inherent ambiguities of opinions due to humor, irony and conversation context. Human conversations often consist of sarcasm and irony that is not easily detected by automated methods and that makes the problem more complex. This work addresses the following question: "Can we develop a knowledge-base of $Cl_A$ and $Cl_B$ related to vaccines and utilize them to identify vaccine-resistance and $Cl_B$ tweets?" The objectives and contributions of the paper are summarized as follows:

- **Knowledge-extraction framework**: To the best of our knowledge, our work is the first work to develop an automatic knowledge extraction architecture to build a knowledge base

---

[1]We use the phrases "vaccine dissenting discourses" and "vaccine dissent" to indicate stances against vaccination. Many phrases, such vaccine hesitancy or vaccine resistance, are used in research studies currently and imply a particular kind of sentiment or position. The word "dissent" captures a range of positions against vaccination, appropriate to the research reported here.

of $Cl_B$ and $Cl_A$ information related to vaccination from web-based sources, and leverage topic-based similarity scoring, agglomerative clustering to build word embedding vectors for $Cl_B$ and correspondingly for $Cl_A$. These word vectors are used to identify $Cl_B$ tweets, and summarize counter-facts based on different categories of $Cl_B$.

- **Identification of types of $Cl_B$ from Twitter discourse**: We develop a novel $Cl_B$ identification technique with very limited labelled tweets to categorize tweets into different sub-classes of $Cl_B$ efficiently. Our module consists of a novel triple-attention based sarcasm detection module that performs well even when the number of labelled tweet samples are limited. Our technique outperforms baselines by a significant margin.
- **Vaccine dissenting discourse analysis**: We present an in-depth discourse analysis using a three-tier knowledge mining module to understand the characteristics of vaccine dissenting users and their tweets as well as their conversational features. These modules have shown promising accuracy in identifying the characteristics of vaccine dissenting discourse, e.g., when more users engage in vaccine dissenting discussion stating $Cl_B$ information, and disapprove vaccination in Twitter.
- Our proposed knowledge extraction and management framework has achieved promising F1-scores, and outperforms baselines by a significant margin ($\approx 14\%$) in identifying $Cl_B$ information in Twitter with limited labelled data and effective analyses of vaccine dissenting discourses.

The rest of the paper is summarized as follows. Section 2 discusses existing works and we present our proposed framework, EVADE in Section 3. The performance evaluation is presented in Section 4, and we conclude in Section 5.

## 2 RELATED WORK

In this section, we briefly discuss related work on vaccination hesitancy and Class-B information propagation in social media.

*Identification of "Vaccination misinformation"[2].* Misinformation detection from online media has made significant progress with high accuracy [1, 2]. However, Depoux, et al., [3] demonstrated that panic created by people on social media spreads fast and therefore such public sentiments, behaviours and rumours need to be detected and responded proactively. Misinformation during COVID-19 outbreak is analysed [4] leveraging the fact-checking platform Tencent from the Chinese social media Weibo. Their work explored that topics, namely, *city lockdown, cures and preventive measures, school reopening, and foreign countries* that evoked the majority of the misinformation. Loomba, et al., [5] conducted a randomized controlled trial in the UK and the USA and quantified how online misinformation on COVID-19 vaccines affects people's intentions with respect to vaccination. The authors also showed that *scientific-sounding* misinformation significantly reduces the vaccination intent among citizens. Another study [6] argued the fact that exposure to misinformation does not necessarily stipulate misinformation adoption. The authors proposed a neural architecture and represented the stances

towards misinformation into a knowledge graph and demonstrated which type of misinformation is mostly adopted or rejected. Most existing works put significant effort in creating new misinformation datasets from Twitter by manual intervention. This practice has significant limitations in terms of coverage and efficacy. Automated misinformation detection methods resort to supervised classifiers, which require substantial number of labelled samples. Given the domain shift from traditional rumour or misinformation detection to COVID-19 vaccination related misinformation, the existing methods fail to provide high-quality results without huge volume of labelled samples. By contrast, we develop an automated knowledge extraction and management framework that can build the knowledge-base from trusted web-sources and can identify misinformation categories leveraging the knowledge base. Our method alleviates the need for manually collecting and curating true facts and labelling efforts. Furthermore, our module can categorize tweets into different vaccination $Cl_B$ subclasses more efficiently and effectively compared to baselines by a significant margin.

*Vaccination Sentiment and Stance Analysis.* Lyu et al. [7] analyzed 40,000 tweets where they manually classified the data into *antivaccine, vaccine-hesitant*, and *provaccine* labels. They utilized multinomial logistic regression and claimed that *socio-economic* factors have a major role in shaping public opinion towards vaccination. Jelodar, et al. used Reddit posts and classified the posts into five sentiment scores using LDA for topic modelling [8], and achieved an accuracy of $\approx 81\%$ on their dataset. Kyle et al. [9] collected COVID-19-Stance data and published using which, the authors trained several stance detection models. Miao, et al., [10] analysed public opinion about lockdown policy in New York State from social media data. Han, et al., [11] explored sentiment analysis in China on COVID-19 and categorized the posts into seven topics, namely "events notification", "popularization of prevention and treatment", "government response", "personal response", "opinion and sentiments", "seeking help", and "making donations". Bechini, et al., [12] proposed a stance detection system to infer stances taken by tweeters from Italy on vaccination. Gupta, et al., [13] presented a framework to mine sentiment of Indian tweeters due to a nationwide lockdown and concluded that the majority of the tweeters supported lockdown. Yu, et al. [14] analyzed the sentiment of COVID-19 related tweets and showed the sentiment distribution across different countries. Unlike existing works, our stance analysis module can identify sarcasm, humour, and irony from Twitter data on vaccination. Our proposed ensemble stance detection module also considers network features such as tweets and posts liked by the user to understand users' sentiments and beliefs. We seek to identify and analyse tweet discourse with "against" stance and "negative" sentiments.

## 3 PROPOSED FRAMEWORK

Figure 1 illustrates the building blocks of the proposed framework, EVADE to identify characteristics of $Cl_B$-leveraging knowledge augmentation and novel information-detection modules.

### 3.1 Pre-processing Module

*3.1.1 Collection and Labelling of Tweet Data.* We used *Twitter streaming API v2 (Academic Research)* to collect tweets in the

---

[2]In this section, we use the term "misinformation" as used in the scholarly work we are referencing, while noting that the definition and scope of misinformation is undefined or varies in many of these.

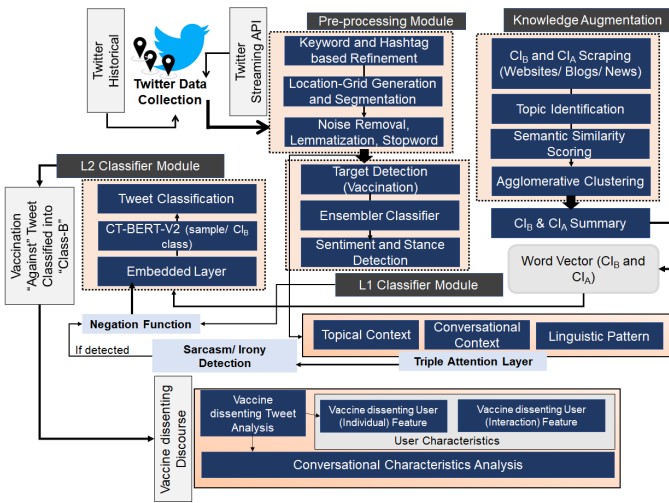

**Figure 1: Overall working modules of proposed knowledge extraction framework (EVADE) for vaccination $Cl_B$ information and vaccine dissenting discourse**

temporal range from October, 2020 to January, 2022 using a *keyword* based search.[3] A *tweet* contains a unique tweet-id *(tId)*, an user-id *(uId)*, text content *(tweet_text)*, timestamp *(t)*, geo-location (co-ordinates of user) *(lat, lng)*, hashtag used *(hashTag)*, number of followers of the user *(no_F)*, number of re-tweets *(no_RT)*, comments *(no_C)* etc. Additionally, we annotated our dataset such that each tweet has three labels: *topic of tweet (tweet_To)*, *sentiment (tweet_S)*, and *stance (tweet_St)* for evaluation. The *tweet_To* can be any of sixteen categories $M1 - M15$ and "Other" (See Table 1). The *tweet_S* has three categories: "positive", "negative" and "neutral"; while *tweet_St* has three categories, namely, "in-favour", "against", "neutral". The geo-location (latitude, longitude) of a tweet is converted to a specific location-string (country, state, city etc.) using *reverse geo-coding* and the *Google Place API*[4]. A *timeline (timeL)* of an event is a sequence of the count of user engagement (tweet, retweet, comment) in that topic in a chronological format. For example, such events may consist of *vaccine unsafe*, or *vaccine can affect fertility*, where the labels are stance (*against*, *in-favour*, and *neutral*) and sentiment (*negative*, *positive* and *neutral)*. The timeL presents the trend of the user-engagement on the event in different stance and sentiment category over a time-period.

Initially, we use a POS tagger to tag each word in *tweet_text*. Next, we perform *Lemmatization* to convert the words to their basic forms using the *WordNetLemmatizer* function of the NLTK python library. For this work, we designated the following as *stopwords*: *'covid19','vaccine','coronavirus','vaccinated', 'vax', 'vaccines', 'covid','vaccination','covid19vaccine'*, and append them with the common stopwords defined in the library.

## 3.2 L1 classifier:Vaccination: In favour +1 | Against -1 | Neutral 0

Our first module (L1 classifier) attempts to classify tweets into three categories: "in favour", "against" and "neutral".

*Stance detection.* Our stance detection module is implemented by ensembling transformer-based pre-trained encoders, namely, $BERT_{LARGE}$, $BERTweet$ [15] and $COVID - Twitter - BERT$ [16]. *COVID-Twitter-BERT* is pre-trained on 97M tweets related to COVID-19. BERTweet is trained using 850M tweets and achieves state-of-the-art benchmarks on both SemEval 2017 [17] sentiment analysis and SemEval 2018 irony detection [18] shared tasks. We selected two BERTweet models (BERTweet-base and BERTweet-covid19-base-cased) and fine-tuned for three downstream tasks: stance detection, sentiment detection and emotion-detection. Hinton, Vinyals, and Dean proposed a *student-teacher* architecture [19] to transfer knowledge from a large teacher model to a small student model by capturing the behaviors of the teacher model. We utilize a *knowledge distillation method* [19] where the teacher model is a self-voted BERT[5], and represented as:

$$L(x, y) = CrE(BERT(x, \vartheta), y) + \chi MSE(BERT(x, \vartheta), \frac{1}{T} \sum_{i=1}^{T} BERT(x, \vartheta_{t-i}))$$
(1)

where BERT(x, $\vartheta$) is the student model, $\chi$ is the weight parameter to balance the importance of two loss functions, namely, mean squared error (*MSE*) and cross-entropy (*CrE*).

However, we propose a different distillation strategy (two-stage fine-tuned strategy) for stance classification. Here, in the first stage, teacher model (pre-trained BERTweet-base on SemEval stance detection dataset) produces stance classes on data (vaccination), which is used as labelled samples to train student models (COVID-Twitter-BERT and BERTweet-covid19-base-cased). In the next stage, ground truth label data (vaccination) is used to fine-tune the student models to achieve better performance as well reducing the overall computational cost.

Another important feature useful for stance detection is the structure of the social networking platform, i.e, social connections and interactions among the users, who voice out their opinion. The above-mentioned distillation method leveraging BERT models attempts to classify stance based on linguistic patterns. However, network features give us strong cues about a person's stance and help us to understand the alignment of a user towards a topic. Connected users influence each other. This work uses two network features: (i) interaction network, where retweets, replies, or any direct mentions are analyzed, and (ii) *preference network* that captures tweets, and comments liked by the users in past seven days. We have considered past seven days data as users' preferences may change over time. Both these features help in stance detection as it captures the users' perceptions and preferences (See second row of Table 6). Next, an *embedding layer* is deployed to augment these two features and refine the final outcome of the stance detection module. We performed a user study to evaluate our system.

*Sentiment detection.* We propose a fusion-model for sentiment analysis of COVID-19 vaccination related tweets. The first layer of the model consists of four classification models: SVM, CNN,

---

[3]The keyword list is present in Appendix A.
[4]https://developers.google.com/maps/documentation/places/web-service/overview

[5]Fine tuning multiple BERT with random seeds, and selecting the output using majority voting.

| ID | Subclasses of $Cl_B$ | Meaning ($Cl_B$) |
|----|----------------------|------------------|
| M1 | vaccine-unsafe-die | Vaccine is unsafe for use |
| M2 | vaccine-substance-development | Contains controversial substances |
| M3 | vaccine-natural-immunity | Natural immunity is better than COVID-19 vaccination immunity |
| M4 | vaccine-makes_me_sick | Vaccine gives you COVID-19, causes variants and other diseases |
| M5 | vaccine-pregancy-fertility | COVID-19 vaccines can make you infertile |
| M6 | vaccine-side-effect | Vaccines contain toxins and cause severe side effects |
| M7 | vaccine-alter-DNA | COVID-19 vaccines interact with human DNA and change it |
| M8 | vaccine-microchip-tracking | The COVID-19 vaccine includes a tracking device |
| M9 | vaccine-not_recommended | Patients with pre-existing health problems are advised not to get the COVID-19 vaccine |
| M10 | vaccine-unnecessary | Pandemic is over and no need to get COVID-19 vaccine shot |
| M11 | vaccine-trust_issue | The effectiveness of vaccinations has never been proven |
| M12 | vaccine-child-infant | The COVID-19 vaccine won't cause severe illness in children, so they don't need it |
| M13 | vaccine-gain-big_companies | Governments and big business are complicit in pushing vaccines despite risks |
| M14 | mask-regulation-not-required | As soon as I get the vaccine, I won't have to wear a mask and taking coronavirus protection measures |
| M15 | vaccine-not_for_me | I'm young & low risk so the COVID-19 vaccine isn't for me |

**Table 1: $Cl_B$ sub-classes, and summarized content by EVADE.**

BiLSTM and COVID-Twitter-BERT. The intuition behind using two types of classifiers (classical and deep learning) is to make the system capable of classifying varied types of test samples. Some studies show [20] that data samples belonging to a low confidence decision region of one classifier may be present in a high confidence decision region of another classifier.

We have adopted a classical support vector machine combined with Bayesian probabilities [21] that uses a Naive Bayes log-count ratio representing the word count feature of the model. We implemented the model using three embedding layers and a sigmoid activation layer. Naïve Bayes log-count ratios are used in the first embedding layer, and the learned coefficients (by SVM) are stored in second layer. Finally, the third layer contains *context specific knowledge* to augment in the model. The context specific knowledge layer represents augmenting *emoticons, emoji, punctuation* of the tweets in sentiment detection. Finally, a dot product is used to make the final prediction. We deploy 1-D convolution with $f$ filter on the input word-embedding matrix $S$. To extract n-gram features, different kernel sizes ($c$) are utilized on the word-embedding matrix at varied granularities (individual sentence and tweet). The feature map is generated by sliding the filter over the complete text: $fm = [fm_1, fm_2, \ldots, fm_{(m-c+1)}]^t \in \mathbb{R}^{(m-c+1) \times 1}$, and the output produced by the convolution is $FM_k \in \mathbb{R}^{(m-c+1) \times f}$. Next, max pooling is used over the feature map to obtain a fixed-size vector, which is then concatenated to form the final representation. The hidden layer of the network is a fully connected layer and finally three softmax cells are used for classification. The hyperparameters used for training are: activation function: ReLu, embedding dimension: 50, number of filters: 150, kernel size: 4, dropout: 0.2, number of neurons in hidden layer: 150], and categorical cross entropy is used as loss function followed by a dropout layer. This work uses BiLSTM as another classifier in the fusion-based model. It uses a bidirectional LSTM for extracting both the preceding and future (sentiment of previous and next unit/ sentence) contexts, the output of the layer is modified as:

$$h_t = \overrightarrow{h_t} + \overleftarrow{h_t} \tag{2}$$

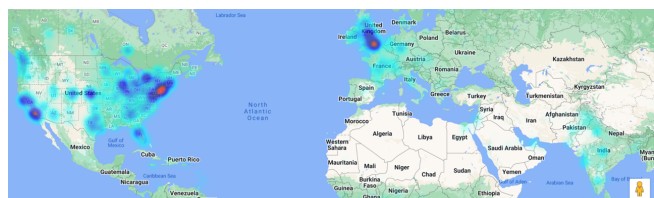

**Figure 2: Distribution of collected geo-tagged tweets**

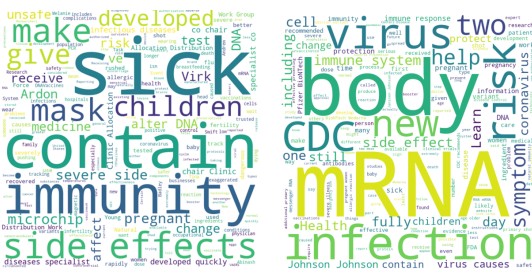

**Figure 3: Wordcloud representing the popular tokens in "Class-B information" (left) and "Class-A Information" (right) category respectively (Count value is represented by font size)**

Here, the output from the forward and backward propagation layer are represented by $\overrightarrow{h_t}$ and $\overleftarrow{h_t}$ respectively. Next, the attention layer is used to measure the importance of several features vectors. we have used the dot product attention function $f_{att}$ and the representation is defined as:

$$r^{att} = \sum_{t=1}^{T} \frac{\exp(f_{att}(h_t, s_t))}{\sum_{i=1}^{T} \exp(f_{att}(h_t, s_t))} h_t \tag{3}$$

The decoder input layer is replaced by the weighted representation ($r_{att}$). Finally, softmax layer is used to get the output labels (sentiment). The network is trained to minimize the cross-entropy loss of the ground truth label and predicted label. The parameters used for training are as follows: embedding dimension: 200, dropout: 0.2,

| Observation & Insights |
|---|
| Initiated tweet (Against) → Conversation thread (majority Against) |
| 21% of the dataset | Users support stating "negative" sentiment about own vaccination experience |
| Initiated tweet (Against) → Conversation thread (majority In Favour ) |
| 26% of the dataset | # of users participating in the thread more compared to # of replies posted by one user | Users posting "positive" sentiment (vaccination experience) and current covid trend |
| Initiated tweet (Class-B information) → Conversation thread (majority $Cl_A$) |
| 11% of the dataset | Major trend observed: Against and $Cl_B$ ($\approx$ 11%) → Sarcasm (Neutral+In favour) ($\approx$ 46%) → In favour ($\approx$ 43%) |
| Initiated tweet (Fact or Class-A information $Cl_A$) → Conversation thread (majority Class-B information) |
| 42% of the dataset | # of tweets from specific users are more compared to # of users in the thread | Several Class-B information classes are discussed in thread | Mentioned "external URLs" or providing references supporting $Cl_B$ for each tweet of $Cl_A$ | Majority of tweets ($\approx$ 76.7%%) contain "mentions" of other users | Majority of tweets ($\approx$ 95.6%) mentioning $Cl_B$ topics show "negative" sentiment about vaccination | Majority of the topics ($\approx$ 83%) include "child vaccine, controversial substance, vaccine makes you sick" |
| Asking for information type tweets → Majority ($\approx$ 87%) replied with "negative" sentiment and Class-B information topics of vaccination |

**Table 2: Conversation sequence analysis**

number of neurons in output layer: 3 activation function: ReLu. Our final model is COVID-Twitter-BERT. We have used the pre-trained model (COVID-Twitter-BERT (CT-BERT) v2 model from hugging face) on 160M tweets between January to July 2020. Finally, all these four base learners need to be fused to train the *meta learner*. We have implemented *stacked generalization* as the fusion method to assign different weights to the output of the base learners (SVM, CNN, BiLSTM, CT-BERT). The fusion method is as follows: (i) The training dataset ($TD$) is split into $N$ equal folds; (ii) Each base learner is applied to all folds excluding one ($TD^{-j}$), and temporary prediction vector is produced, (iii) Next, new training dataset ($TD'$) is used by augmenting the temporary predictions to train the meta learner. It may be noted that to yield better efficacy, base learners must have lower classification error. This work selects the base learners considering this. Finally, iterative gradient boosting algorithm is deployed to create the final fusion outcome.

### 3.3 L2 classifier: Categorize "Against" tweets into Class-B information classes

The next module is *L2 classifier* which categorizes the "against" and "negative" sentiment tweets into sixteen Class-B information classes (M1-M15 and Other, See Table 1). This task is divided into following sub-modules.

*3.3.1 Building knowledge-base from trusted sources.* In this section, we present the automatic knowledge extraction and augmentation method to alleviate the need of ample amount of labelled tweets of different $Cl_B$ categories. Moreover, it might be noted that $Cl_B$ types may change over time, therefore devising supervised training based on labelled tweets is not a feasible option as well. In this regards, we aim to build our knowledge-base from automatic scraping of trusted sources as illustrated in Figure 4.

- We develop a crawler which scrapes information from websites, blogs and news-articles where $Cl_B$ and facts ($Cl_A$) about COVID-19 vaccination are specifically mentioned. We have also considered different opinions such as *vaccine dissenting* and *pro-vaccination* webpages to build the knowledge-base. To implement the crawler scipt, we have used *beautifulsoup4*[6] python library for parsing HTML and XML data.

**Figure 4: Sources of knowledge-base of $Cl_B$ of vaccination**

The script searches for words "Misinformation", "Myths", "Truth", "Fact", and identifies the intermediate blocks of text within two such words. For parsing PDFs (since few web-links contain PDFs), we have used *Pytesseract*[7] python library which is a OCR tool. As the crawler script scraps and creates "$Cl_B$" and corresponding "$Cl_A$" dataframes automatically without manual intervention, we can append more sources at any time of the development process making the knowledge extraction pipeline flexible.

- As of now, we have scraped 80 sources including webpages, blogs and news articles and collected 488 $Cl_B$ and corresponding counter $Cl_A$. However, given the information is amassed from different sources, it has repetitive data making the knowledge base redundant. To resolve this issue, next we devise a semantic scoring mechanism and clustering to extract unique $Cl_B$ information categories.

*3.3.2 Clustering.* We devised sentence (each $Cl_B$) embedding on semantic similarity to cluster similar type of $Cl_B$. Here, we have adapted pre-trained t-BERT, 2020 model [31] for sentence embedding. A variant of *t-BERT (topic-informed BERT-based architecture)* is used for pairwise semantic similarity detection. Here, we have used two categories (Class-B and Class-A) in the architecture

---

[6]https://beautiful-soup-4.readthedocs.io/en/latest/

[7]https://pypi.org/project/pytesseract/

to infer similarity between both "$Cl_B$" and "fact ($Cl_A$)" per class (M1-M15, See Table 1). Next we have devised *Agglomerative clustering* on the similarity score matrix values to cluster similar $Cl_B$ into same categories. Each of the classes contains similar $Cl_B$ and their countering scientific-consensus-based facts. In this method, we have automatically extracted 15 $Cl_B$ classes as represented in Table 1. We summarize the corresponding $Cl_A$ to each $Cl_B$ class of the knowledge-base using T-BertSum [32]. This can be utilized to recommend according to predicted tweet class as de-escalation strategy of $Cl_B$ propagation and providing correct information ($Cl_A$) to vaccine resistant people.

### 3.3.3 Tweet Classification on Class-B information ($Cl_B$) categories.
We have constructed the word embedding vectors of $Cl_B$ and $Cl_A$ classes derived from the previous step.

- Each $Cl_B$ class has a word embedding vector obtained from fine-tuning *BERTopic* [33] embedding layer, namely *em_vec*. Contrary to document embedding using BERTopic, we feed all text data of each $Cl_B$[8] in the pre-trained language model and extract topic-representations. We skip the second step of BERTopic which clusters the embeddings of the conventional document embedding, as our input is already clustered based on domain-specific (COVID vaccination) knowledge
- CT-BERT V2 is used for L2 classifier, where we added two layers (layer 0, layer 1)
- Layer 0 of CT-BERT V2 is trained using *em_vec* which helps to augment coherent topic representations for each $Cl_B$
- Additional embedding layer (Layer 1) is deployed using labelled tweets (#100) of each category of $Cl_B$ (M1-M15) which helps in further fine-tuning the L2 classifier.

We analyzed incorrect test samples from L2 classifier, and observed classification errors due to different factors as mentioned below:

- *Sarcasm/ irony (contributing ≈ 73% of the error samples)*: For example " I got my microchip . . . I mean my first dose of the Covid vaccine today. Have I turned into a zombie or vampire" [Model predicted it as "against" and $Cl_B$ class M8] "hope the covid vaccine alters my dna and I get to join the x men" [Model predicted it as "against" and $Cl_B$ class M7] "A new strain, more contagious . . . yet the same rushed vaccine will save you? Hurry up and get in line for your shot!!!" [Model predicted it as "in favour"]
- *Asking for information (contributing ≈ 21% of error samples):* User is requesting for more information for deciding regarding vaccination shot. For example: "I am cancer survivor. Is it unsafe for me to get the vaccine? Whether I am higher risk of developing serious sideeffects from the shot?" [Model predicted as "against" category and $Cl_B$ class M1] "My kids are turning 8 soon. Will more dosage mean better longer lasting immunity or severe sideeffects? Child COVID vaccine battle heats up in Sacramento. Is mandating it for all kids premature?" [Model predicted as "against" category and $Cl_B$ class M12]
- *Incomplete/ Out-of-context (Contributing ≈ 6% of the error samples):* This category includes either out-of-context tweet

samples or incomplete tweets where proposed model fails to detect the context of the tweet. For example: "If the vaccine is to help with depopulation, what does the actual virus help with?" [Model predicted as "neutral"]

We propose a triple-attention based model for identifying sarcasm and refining the categories by considering above-mentioned error classes and enhancing the accuracy. It may be noted that existing approaches fail to identify such scenarios effectively: (a) Supervised technique where sarcasm detection model is trained using common texts from wiki and sarcastic similie does not work for our scenario due to discourse domain shift to COVID-19 and vaccination topics. (b) Hashtag based refinement does not work as the tweet samples do not contain specific hashtags such as #*sarcasm*, #*sarcastic*, or sentiment based #*sad*, #*excited*. (c) Rule based approach is not suitable either due to the requirement of large sarcasm-labeled corpus (on COVID vaccination). Our aim is to identify such sarcasm or irony from Twitter discourse with limited labelled data (COVID-19 vaccination). The triple-attention based layers are mentioned as follows:

- Layer 1: Topical Context - Some topics are more prone to sarcasm than others. For example, tweets about controversial topics like microchips, DNA changes, etc. are more likely to draw sarcasm than tweets about vaccine side effects. Here, we implemented *LDA* for topic modelling controversial topics and classifying sentiment of Tweets into "Positive", "Negative", "Sarcastic". This layer has a fully connected self-attention layer.
- Layer 2: Conversational Context - It refers to text in the conversation of which the target tweet is a part. We considered "Re-tweet (original tweet stance analysis)" and "Replies in the thread" to understand the conversation context of the tweet. Target tweet and previous tweet in the conversation thread are analysed along with comments in thread structure. Further, a sequence labelling (positive, negative, sarcastic) of the tweets in the sequence is done to predict sarcasm in every text unit in the sequence.
- Linguistic Pattern Discovery: Sarcasm can be detected by the contrast between positive verbs and phrases indicating negative situations [e.g. "Oh sure! I support untested and unverified vaccine. Lord save the youth!"]. Here, we identify contexts that contain a positive sentiment contrasted with a negative situation [OR negative sentiment contrasted with a positive situation]. We have devised an iterative training step: Take "seed word" (e.g. support, save, rush) and sarcastic tweets and extracting phrases having contrasting polarity. This information is used to obtain embedding vector from different seeds.
- Features used: (i) Sentiment incongruities: The frequency with which a positive word is followed by a negative word and vice versa), (ii) Largest positive/negative subsequence: The length of the longest series of contiguous positive/negative words, and (iii) Pragmatic features: Existence of emoticons, laughter expressions, punctuation marks, ellipsis and capital words.

Using triple-attention layer, each of the tweets is classified as "sarcasm (yes)" or "sarcasm (no)". Next, a "negation function" is used

---

[8]After clustering, each $Cl_B$ class has several similar items obtained from different web-sources

on the output of L1 classifier, which means if sarcasm is detected and L1 predicted class is "against", then it is marked as "in favour", and vice-versa. If L1 classifier output is "neutral", then the sentiment of the tweet is verified, and assigned accordingly. Finally, "against" tweets are passed into L2 classifier for identifying $Cl_B$ classes. For identifying "asking for information" type of tweets (See section 3.3.3), we have used pragmatic feature of the tweets, namely, (a) identification of punctuation mark (?) and *wh-word*; (b) inspecting whether the polarity of the tweet as "neutral". The issue of the *incomplete/ out-of-context* tweets are resolved by "conversational context" layer of triple-attention model, and filtering tweets using "length" based constraints (we select tweets having atleast 50 length (characters) excluding URLs).

## 3.4 Vaccine dissenting Discourse Analysis

In this section, we analyse vaccine dissenting discourse in Twitter considering *vaccine dissenting tweet* analysis and *vaccine dissenting user* analysis. For vaccine dissenting user analysis, we have used 380 users' dataset [34].

- For vaccine dissenting tweet analysis, following features are obtained: (a) Text pattern by analyzing sentence types, use of determinants, special characters, and modifiers (b) Text readability metrics: word structure, average syllables per word, easy word use ratio in a word list, and sentence complexity (c) Textual perception and informative opinion based on semantics and subjectivity (d) Speech information parts: number of verbs, adjectives, adverbs, syllables, and words, rate of adjectives, adverbs, and words per sentence (e) Capitalization features: words with initial caps and all caps and the number of POS tags with at least initial caps and (f) Word unigrams/ bigrams: Cluster words used in the similar contexts
- Vaccine dissenting users' characteristics analysis is performed based on *individual user feature* and *communication based feature* as follows.
- Following features are obtained for vaccine dissenting user analysis (Individual): (a) Historical topics: Topic-based features by inferring a user's 100 topics over all tweets (initiated) (b) Profile information: count of friends, followers and statuses, duration on Twitter, average number of posts per day, location (if available), gender (if available), and verified by Twitter (c) Historical sentiment: Distribution over sentiment in the user's historical tweets (d) Interactional topics: Topic-based features where user interacts (re-tweet, quote-tweet, comment/ reply)
- Communication based features are as follows where more than one user interacts: (a) Degree of interaction between two users: count of previous messages sent from the author to the addressee and (b) Rank of the addressee among the user's -mention recipients.

For detecting vaccine dissenting user, we have deployed *Gradient Boosted Decision Trees (GBDT)*, an ensemble of decision trees using the above-mentioned feature sets. It is fitted in a forward step-wise manner to current residuals of the decision nodes.

| Observations | Vaccine dissenting (Yes) | Vaccine dissenting (No) |
|---|---|---|
| Avg tweet length | 196.2 | 107.8 |
| Avg tweet & comments posted per day | $\approx 26$ | $\approx 22$ |
| Avg tweet & comments posted/ day (Vaccination) | $\approx 21$ | $\approx 4$ |
| Re-tweet count | 18.45 | 4.61 |

**Table 3: Observed trends on vaccine dissenting user and tweet feature analysis**

## 4 PERFORMANCE EVALUATION

To evaluate the efficacy of our data analytics framework, we have used one public dataset [34] containing 6-months dataset of $\approx 380$ vaccine dissenting users' tweets, and our collected dataset of $\approx 1.5M$ tweets in October 2020 - January 2022 time-span. Figure 2 shows the data distributions of our dataset on a spatial scale[9].

### Accuracy: Vaccination stance detection and $Cl_B$ classification

The accuracy of sentiment classifier module is shown in Table 5 where our sentiment classifier model has achieved $\approx 83\%$ accuracy in classifying the sentiments of the tweets. Comparison has been carried out with eleven classifier models. Amongst the classical models, SVM outperforms others, and therefore selected as one of the base learners. The key reason of the better accuracy is EVADE's fusion-based method using four classifiers which demonstrate better accuracy compared to other baselines. Vaccination stance detection accuracy is reported along with an ablation study in Table 6 to illustrate the impact of different layers of L1 classifier. It is observed that our proposition of augmenting network features (second row) and triple attention layer have boosted the performance significantly, specifically for "against" vaccination stance. Table 4 shows the performance of $Cl_B$ classification (L2 classifier) in terms of F1-score for the fifteen different targets or $Cl_B$ topics. We have also reported the accuracy of other three BERT models to demonstrate the usefulness of our ensemble method. Our framework has outperformed other BERT models for most of the targets, and yields 92.48% and 87.18% accuracy for different $Cl_B$ classes respectively, which is quite promising given the complexity of the problem. Figure 3 illustrates the word cloud of topics used in both $Cl_B$ and $Cl_A$ in Twitter.

### Vaccine dissenting discourse analysis insight

The vaccine dissenting discourse analysis helps in identifying vaccine dissenting users and predicting a thread into vaccine dissenting discourse. Table 3 represents the overall trends of vaccine dissenting users compared to a vaccine non-dissenting (or common user). It is observed that average tweets and comments posted by vaccine dissenting and common user is comparatively similar, however, vaccine dissenting users post 0.8 times more than common users regarding vaccination topics. The average re-tweet, favourite and followers count of vaccine dissenting user profiles are significantly higher than common users. Our vaccine dissenting user classifier produces 0.896

---

[9]This shows only the geo-tagged tweets of our collected dataset. Though, the data analysis and evaluation have been done irrespective of geotagging.

| Model | M1 | M2 | M3 | M4 | M5 | M6 | M7 | M8 | M9 | M10 | M11 | M12 | M13 | M14 | M15 |
|---|---|---|---|---|---|---|---|---|---|---|---|---|---|---|---|
| $BERT_{LARGE}$ | 78.2 | 74.6 | 80.5 | 82.7 | 78.03 | 75.81 | 82.08 | 80.24 | 84.10 | 81.04 | 80.10 | 80.09 | 82.17 | 80.94 | 78.16 |
| COVID-Twitter-BERT | 84.08 | 89.12 | 87.45 | 83.10 | 82.08 | 83.11 | 85.18 | 82.06 | 76.20 | 78.18 | 81.90 | 82.01 | 83.98 | 83.43 | 84.07 |
| BERTweet-covid19-base-cased | 83.02 | 85.11 | 80.83 | 83.18 | 81.23 | 83.02 | 84.16 | 83.07 | 86.15 | 83.04 | 81.05 | 82.7 | 81.44 | 81.73 | 82.19 |
| **EVADE** (Proposed) | **92.48** | **90.04** | **88.02** | **89.16** | **88.04** | **89.90** | **92.01** | **90.05** | **87.18** | **89.03** | **91.45** | **88.02** | **87.94** | **91.80** | **88.43** |

**Table 4: Comparison of accuracy (F1-score) of L2 module with baselines for categorizing tweets into $Cl_B$ classes.**

| Classifier | Positive | | Negative | |
|---|---|---|---|---|
| | Precision | F1-score | Precision | F1-score |
| SVM | 0.68 ± 0.002 | 0.65 ± 0.006 | 0.624 ± 0.012 | 0.608 ± 0.005 |
| Random Forest | 0.67 ± 0.005 | 0.642 ± 0.002 | 0.619 ± 0.011 | 0.582 ± 0.002 |
| KNN | 0.545 ± 0.005 | 0.528 ± 0.011 | 0.491 ± 0.011 | 0.462 ± 0.004 |
| XG Boost | 0.660 ± 0.004 | 0.643 ± 0.011 | 0.601 ± 0.004 | 0.570 ± 0.006 |
| Gaussian Naïve Bayes | 0.562 ± 0.006 | 0.541 ± 0.010 | 0.510 ± 0.014 | 0.508 ± 0.006 |
| AdaBoost | 0.631 ± 0.005 | 0.618 ± 0.014 | 0.584 ± 0.012 | 0.540 ± 0.010 |
| Perceptron | 0.668 ± 0.010 | 0.640 ± 0.006 | 0.603 ± 0.010 | 0.577 ± 0.005 |
| LSTM | 0.725 ± 0.005 | 0.713 ± 0.005 | 0.709 ± 0.010 | 0.701 ± 0.019 |
| BiLSTM | 0.759 ± 0.023 | 0.748 ± 0.045 | 0.712 ± 0.012 | 0.708 ± 0.016 |
| $BERT_{BASE}$ | 0.825 ± 0.002 | 0.79 ± 0.012 | 0.809 ± 0.0062 | 0.77 ± 0.003 |
| $BERT_{LARGE}$ | 0.836 ± 0.017 | 0.810 ± 0.005 | 0.81 ± 0.011 | 0.792 ± 0.016 |
| **EVADE** (Proposed) | 0.842 ± 0.006 | 0.818 ± 0.003 | 0.835 ± 0.02 | 0.812 ± 0.010 |

**Table 5: Comparison on sentiment analysis classifier.**

| Model | Against | | | In Favour | | |
|---|---|---|---|---|---|---|
| | Precision | Recall | F1-score | Precision | Recall | F1-score |
| L1 (linguistic) | 0.742 | 0.816 | 0.77 | 0.818 | 0.850 | 0.833 |
| L1+network | 0.765 | 0.847 | 0.8039 | 0.826 | 0.851 | 0.8383 |
| L1+topical | 0.78 | 0.848 | 0.8125 | 0.828 | 0.853 | 0.840 |
| L1+conversational | 0.793 | 0.851 | 0.8209 | 0.846 | 0.852 | 0.848 |
| L1+Linguistic (Sarcasm) | 0.801 | 0.845 | 0.822 | 0.853 | 0.861 | 0.856 |
| L1+triple attention (all) | 0.886 | 0.854 | 0.869 | 0.87 | 0.864 | 0.866 |
| **L1+FULL** | **0.914** | **0.8537** | **0.882** | **0.881** | **0.872** | **0.876** |

**Table 6: Ablation study on L1 classifier module**

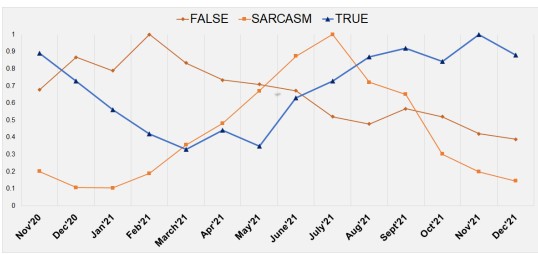

**Figure 5: Timeline of Tweets (counts – normalized into 1-0 range based on the predicted data samples in each category) on "False", "Sarcasm" and "True" information regarding vaccination from Nov 2020 – Dec 2021**

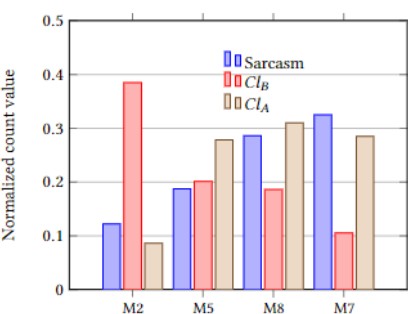

**Figure 6: Normalized value representing four topics which evoke maximum sarcasm**

F1-score to classify users into "vaccine dissenting (yes)" or "vaccine dissenting (no)".

Table 2 represents the observations of conversation sequence analysis of vaccination discourse in Twitter. We have identified four scenarios from the dataset: (i) when the initiated tweet of a conversation thread is *against* vaccination and majority of the comments are *against* in the thread; (ii) when the initiated tweet of a conversation thread is *against* vaccination, however, majority of the users in the discourse are *in favour* of vaccination and disapproved the initial tweet. In this case, we observed that users are sharing vaccination experience with "positive" sentiment and countering the vaccine dissenting users. (iii) when the initiated tweet presents "$Cl_B$" and majority of commenters in the discourse disapproved the topic. In general, we observed an interesting trend, where participants in the thread initially supported the tweet in a sarcastic way and then finally disapproved the topic. (iv) when initiated tweet states true information about vaccination, however majority of the comments present $Cl_B$: Here, we observed specific characteristics of the discourse, where users mentioned several $Cl_B$ topics, and three most mentioned $Cl_B$ classes are M12, M2 and M4. Further, a large amount of external links, references and mentions are observed in this discourse. Figure 6 illustrates top four topics which evoked sarcasm on Twitter discourse along with the normalized count (e.g., the first blue bar denotes the ratio of tweets in M2 category representing sarcasm and

tweets in all categories representing sarcasm) of "sarcasm", $Cl_B$ and $Cl_A$ tweets in these topics. Figure 5 illustrates the tweets presenting $Cl_B$, sarcasm and true information regarding vaccination in the time-range November 2020 to December 2021.

## 5 CONCLUSION

In this work, we show how to develop knowledge base and augment the knowledge to classify tweets into different $Cl_B$ classes proposing knowledge extraction and tweet discourse analytics modules. Our proposed framework, EVADE is useful for efficient stance analysis towards vaccination, $Cl_B$ detection and integration of external knowledge (scientific facts about vaccination from trusted source) to paint a comprehensive picture of information extracted from social media data such as tweets. Our automated data analytics framework helps understand public opinion regarding COVID-19 vaccination and related $Cl_B$ topics. Further studies can emphasize the analysis of social network topology to detect echo-chamber effects about

vaccine dissent leveraging our knowledge base. We will also explore vaccination dissenting discourse on different vaccine types as an extension of our present work. We strongly believe that our present work will act as a foundation for developing advanced knowledge extraction models to perform complex semantics mining tasks in social media domain.

**Acknowledgement** We thank Lavan Bathija and Asaiil Aksari of College of Information Sciences and Technology, Penn State for helping in data curation and annotation.

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

# A    KEYWORD LIST

unsafevaccine , bloodclotcovidvax , saynotovaccine , justsaynovaccine , headchevaccine , covidchestpainseconddose , vaccinesideeffect , safetyissuevaccine , humantrialfailcovid19 , vaccinedanger , persistentabdominalpaincovid19, persistentabdominalpaincovid19 seconddose , heartandrespiratoryconditionsvaccine , WHOLiedPeopleDied , trumliedpeopledied , chinaliedpeopledied , trustpublicfigurevaccine , showdatavaccinemanufacturer , distrustgovernment , distrusthealthorganizations , influencevaccine , bidenadministrationlossvaccine , trustpresidentvaccine , approvedvaccinenoproof , implicationsforpublichealthpractice , firstvaccineapproved , noassurancenewvaccine , notrialapprovedvaccine , nodataapprovedvaccine , adverseeffectvaccine , compliancevaccinedata , quickdevelopmentvaccine , noqualitystandardcheckingvaccine , protectinfantagainstvaccine , protectchildagainstvaccine , infantmortalityvaccine , immatureimmunesystemchild , imunesystemoverwhelmchildprotection , toomanyvaccineinfant , disorderimmunesysteminfantvaccine , stopchildtrialvaccination , Herdimmunitycovidvaccination , Naturalimmunityworksbetter , Vaccinedoesnotboostimmunity , NaturalImmunityIsBetterThanVaccineacquiredImmunity , Infectioninducedimmunitybetterthanvax , novaccinenodeath , naturalinfectionbetterworks , Regularhygieneworksbetter , Protecthealthbysanitationnovax , Personalhygienepreventsinfectionnovaccine , Safewashservicenocovid , Betterpublichealthpreventscovidnotvaccine , Investcorepublichealthinfrastructurenotinvaccination , Washyourhands , Simplehygienemeasureprotectsyounotvaccine , Vaccinesleadlongtermeffects ,

Shreya Ghosh[1], Prasenjit Mitra[2], Bernice L. Hausman[3]

vaccinescausecancer , covidvaccineinfectsyou , covidvaccinecausecomplication , covidvaccinethreats , covidvaccinealterdna , vaccinesmakeyouvulnerablelongtermillness , vaccinationleadstohospitalization , vaccinemakesmychildsick , vaccineeffectsmoreinfant , vaccineleadstoinfectioninfant , vaccinegivesunusualreactionchild , childrenathigherrisknovaccine , saveyouchildlifenovaccine , vaccineexposesmychildtocovid , Herdimmunityendofcovid , flatteningcurvenovaccine , zerocasesnovaccinenocovid , covidcasesdropnovaccine , finaldestinationherdimmunity , nocovidnovaccinemandates , coronavirus, corona virus, Coronavid19, coronavirususa, coronavirusaustralia, covid19, covid-19, covid-19, coronavirus, coronapocalypse, quarantinelife, socialdistancing, SocialDistancing, StayHome, StayAtHome, lockdown, StayHomeSaveLives, Quarantine, socialdistancing, confinement, FlattenTheCurve, StayHomeStaySafe, stayhome, QuarantineLife, 5G, TrumpVirus, StaySafe, Coronavirustruth, WashYourHands, ChineseVirus, TrumpLiedPeopleDied, stayhome, Lockdown, TrumpLiesAboutCoronavirus, ChinaVirus, COVIDIOTS, COVIDIOT, quarantinelife, StaySafeStayHome, hoax, TrumpVirusCoverup, panicbuying, Hydroxychloroquine, TheLockdown, lockdowneffect, toiletpaper, StayAtHomeAndStaySafe, Stay TheHome, SelfIsolation, QuarantineAndChill, stayathome, TrumpPandemic, SocialDistanacing, ChinaLiedPeopleDied, QuaratineLife, lockdownextension, Trumpdemic, TrumpLiedPeopleDied, WorkFromHome, TrumpLiesPeopleDie, QuarentineLife, TrumpLiesAmericansDie, Lockdown21, workingfromhome, TrumpOwnsEveryDeath, TrumpPlague, LockdownExtended, CoronavirusLockdown, Trump Genocide, SocialDistancingNow, CCPVirus, SocialDistance, ChineseVirus19, ShelterInPlace, StayAtHomeSaveLives, PhysicalDistancing, Resist, Isolation, ChinaCoronaVirus, toiletpapercrisis, lockdownuk, chloroquine, WFH, ChinaLiedAndPeopleDied, LockdownNow, selfisolating, Lockdownextention, CloseTheSchools, Pencedemic, SupportLockdownStaySafe, toiletpaperpanic, schoolclosure, ToiletPaperApocalypse, selfquarantine, masks, handwashing, WearAMask, SafeHands, handsanitizer, LockDown, mask, isolation, flattenthecurve, washyourhands, panicbuyers, panickbuying, Social Distancing, ChinaMustExplain, Masks4All, WashYourHandsChallenge, BloodOnTrumpsHands, IsolationLife, Hoax, ToiletPaperPanic, toiletpapergate, homeschooling, panicshopping, 5GKILLS, hydroxychloroquine, LockdownHouseParty, trumpvirus, StayHomeSaveLifes, homeoffice, PencePandemic, FamiliesFirst, StayHomeCanada, facemasks, selfisolation, flatteningthecurve, QuaratineAndChill, HerdImmunity, AloneTogether, Hydroxycloroquine, workfromhome, remotework, Masks, FlattenTheCuve, COVIDIDIOT, Socialdistancing, hydroxychloriquine, day8oflockdown, wfh, stayHome, herdimmunity, CoronavirusLockdownUK, TrumpVirus2020, TrumpBurialPits, ShutItDown, 5GCoronavirus, Homeoffice, Resistance, ChineseVirusCorona, chinesevirus, panicbuyinguk, KungFlu, NYCLockdown, facemask, trumpandemic, CoronaHoax, HomeOffice, ChineseCoronavirus, Pandumbic, CoronaLockdown, OPENAMERICANOW, TogetherAtHome, testing, FeverDetectionCamera, WhereAreTheTests, vaccines, Plandemic, Scamdemic, FireFauci, StudentLivesMatter, StayatHome, endthelockdown, ReopenAmerica, lockdown2020, CancelAPExamsPromoteStudents, schoolreopening, HealthOverExams, PromoteStudentsSaveFuture, TestingTestingTesting, schools, lockdownUKnow, SaferAtHome, ContactTracing, FreeThemAll, TrumpCoronavirusTestFailure, TrumpLiedAmericansDied, Handwashing, ChinaLiedPeopleDie, StayAtHomeOrder, OpenAmerica, Vaccine, remoteworking, californialockdown, TestTraceIsolate, EndTheShutdown, WHOLiedPeopleDied, Curfew, ReOpenAmerica, TestingVIRUSNOW, socialdistance, pandemic, FakePandemic, stayhomestaysafe, TrumpPandemicFailure, BackToWork, chinavirus, ReopenAmericaNow, MakeChinaPay, TestAndTrace, MasksOff, SayNoToMasks, ConstitutionOverCoronavirus, endthelockdownuk, StudentBan, SchoolsMustOpeninFall, SchoolReopening, Hydroxychloroquine