# OpenReview forum: "EVADE: Exploring Vaccine Dissenting Discourse on Twitter"
_ACM.org/SIGKDD/2022/Workshop/epiDAMIK — KDD 2022 Workshop epiDAMIK Oral_

### Official Review · Reviewer_YDhQ · 2022-06-23
**The authors of the paper "EVADE: Exploring Vaccine Dissenting Discourse on Twitter" have provided the Knowledge extraction methods to extract public opinion on covid-19 vaccines in a clear and concise manner. In this study, the authors curate their own data using Twitter and construct two innovative classfiction models, L1 and L2, that use the SOTA transformer model and the fusion process.**

**Rating:** 4
**Confidence:** 3

**Review:**

Pros:
1) Misinformation about the covid-19 vaccination is the greatest and most difficult challenge, and social media is widely recognised as the epicenter of misinformation spread. I am thrilled to see that the author has selected a vital and difficult job.
2) The authors have developed an intriguing way of extracting information from social media data.
3) The writing quality of the paper is really clear and succinct.

Cons:
1) Authors must segment their analyses for all available vaccines.
2) Details of model training are insufficient
3) Why have authors performed Vaccine dissenting Discourse analysis on distinct datasets?


Despite these concerns, I feel that authors should be appreciative of the quality of their work. I accept the paper in its present format.

---

### Official Review · Reviewer_HkB5 · 2022-06-25
**Interesting work on categorizing covid vaccine-related tweets**

**Rating:** 4
**Confidence:** 4

**Review:**

This paper works to categorize tweets about covid vaccines with an ultimate aim to detect tweets about vaccines that are not backed by science. The authors present an extensive architecture for categorizing tweets, which seems to be successful in the experiments.

This paper has many notable strengths:
- The architecture is extensive and incorporates intuitive components of the key problem: Detecting in-favor or not-in-favor vaccine tweets, detecting not-in-favor tweets that are also scientifically unfounded, and digging into each section to categorize further.
- The motivation for this work is strong and of-interest to this workshop and may spur interesting conversations.
- There are lots of experiments that assess this work from many angles.
- Since there are so many moving parts in the method, it is nice to see an ablation study (more focus on this would be helpful to the reader, still)

This paper could be improved with the following considerations:
- It would help in the intro to position this work in the context of the literature: Is Cl_A vs CL_B a standard taxonomy from the literature?
- Employing the notation "$\text{Cl}_B$" obfuscates the narrative via arcane diction. Maybe you could say the tweets have "unfounded claims?" It's hard to remember the nuance of CL_A vs. CL_B through the whole paper.
- Figure 3 isn't mentioned in the text and word clouds rarely help the story. In this case it doesn't seem to help the reader.


Minor Points:
- Line 139 should begin with "The rest..."
- Table 3 and Figure 2 run into the margin

---

### Official Review · Reviewer_qf8C · 2022-06-27
**Successful framework for vaccine discourse analysis, scope for improving presentation**

**Rating:** 5
**Confidence:** 4

**Review:**

The paper proposes a framework for knowledge extraction, categorization, and understanding of public discourse around vaccines during COVID-19. They have an "automatic" knowledge extraction framework that further performs the categorization of tweets without needing large amounts of labeled data. External reliable webpages are used to create a knowledge base that assists the classification. Additionally, the framework has various modules for pre-processing, a high-level classifier (stance detection - in favor, neutral, against), sarcasm detection, and sub-classification within "against" to identify the reason for vaccine dissent. The experiments demonstrate significant improvements over the baseline.

Pros:
- Many components in the framework are successfully integrated -- performance is significantly better than the baseline.
- Uses external knowledge to compensate for lack of labeled data.

Cons:
- Presentation can be significantly improved (see below).

My main concerns are the following:

- It is difficult to judge if the individual components of the framework are novel. For example, there has been a lot of work in sarcasm detection. It is not clear, whether the proposed method is novel or borrowed, or motivated by existing work.

- Due to the presence of multiple components, many modeling decisions had to be made. The paper does not defend these choices. One may compare the chosen design against other potential methods. It is understandable that ablation studies with so many components may be difficult. While the choices are reasonable, the authors should at least present the motivations behind their choices. For example, why was "triple attention" used? Why not a series of dense layers?